# Microstructure and Mechanical Properties of Porous NiTi Alloy Prepared by Integration of Gel-Casting and Microwave Sintering

**DOI:** 10.3390/ma15207331

**Published:** 2022-10-20

**Authors:** Zhiqiang He, Ze Wang, Dezhi Wang, Xinli Liu, Bohua Duan

**Affiliations:** School of Materials Science and Engineering, Central South University, Changsha 410083, China

**Keywords:** gel-casting, porous NiTi alloy, microwave sintering, microstructure, mechanical properties

## Abstract

Porous NiTi alloys were manufactured by integration of gel-casting and microwave sintering. The effects of sintering temperature on porosity, compressive strength, pore morphology and phase composition of sintered samples were researched. The results show that the porosity and the mean pore diameter of porous NiTi alloys decrease with increasing sintering temperature, whereas the content of the NiTi phase, the elastic modulus and compressive strength of sintered samples increase. When the gel body with the solid loading of 50 vol.% is microwave sintered at 1000 °C for 30 min, porous NiTi alloys are obtained with the porosity of 38.9%, the compressive strength of 254 MPa, elastic modulus of 4 GPa, and predominant phase of NiTi. The results suggest that the method is suitable for rapid preparation of large-size and complex-shape personalized products similar to human bones at a low cost.

## 1. Introduction

Porous NiTi alloy has attracted much attention as a promising biomaterial for biomedical applications, such as orthopedic implants and hard-tissue replacement, since it not only retains the superior performance of bulk NiTi alloy including pseudoelasticity, excellent corrosion resistance and biocompatibility, but also possesses an interconnected pore structure which is helpful to body fluid transmission, bone tissue in-growth and elastic modulus adjustment [1,2,3,4]. With the aging of the population and frequent traffic accidents, the market demand for porous nickel titanium parts has increased dramatically. Most of these parts are complex in shape and require personalized design, so a low-cost and rapid preparation method is urgently needed.

At present, there are two type of preparation methods of porous NiTi alloy, namely additive manufacturing (AM) and conventional powder metallurgy (PM). Conventional PM comprises some processes such as conventional sintering, hot isostatic pressing, spark plasma sintering, and self-propagating high-temperature synthesis [5,6,7,8,9,10]. Nevertheless, these processes make it difficult to directly prepare products with complex shapes, and are time-consuming. As a novel sintering technique, microwave sintering exhibits some advantages, such as rapid heating rates, reduced sintering times and energy consumption. In consequence, it has been successfully applied to the preparation of porous NiTi alloy [11,12,13]. However, it still cannot fabricate parts with complex shapes. In contrast, some AM processes such as selective laser sintering, laser engineered net shaping, and selective laser melting can easily solve this problem, but suffer from high cost and expensive equipment [14,15,16,17]. Several low-cost near-net-shape forming technologies including metal injection molding (MIM) and gel-casting have been introduced [18,19,20,21]. Compared with MIM, gel-casting not only overcomes its drawbacks such as the long debinding period and the high binder requirement, but also is particularly suitable technology for fabricating products with large size and complex shapes at a small scale [22,23,24].

In the previous work, we prepared porous NiTi alloy using a gel-casting method followed by conventional sintering. However, the whole preparation period was still long due to the use of conventional densification measures [20,21]. The aim of this work was to combine the advantages of gel-casting technology and the microwave sintering process to fabricate a porous NiTi alloy.

## 2. Experimental Procedure

### 2.1. Raw Powder and Reagent

Hydrogenated dehydrogenated titanium powder (purity of 99.5%, mean diameter of 6.37 μm) and atomized nickel powder (purity of 99.5%, −325 mesh) were selected as the raw powders. The morphologies of the raw powders are shown in Figure 1. Ti powder has an irregular polygon morphology, while Ni powder is spherical which is conducive to obtaining a slurry.

The reagents used in the gel system were as follows. N-octanol, hydroxyethyl methacrylate (HEMA), 1,6-hexanediol diacrylate (HDDA), tert-butyl peroxybenzoate (TBPB) and N,N-dimethylaniline (DMA) were used as solvent, organic monomer, crosslinker, initiator and catalyst, respectively, and they were all supplied by Aladdin Industrial Co., Ltd. Silok-7456F from Shenzhen Silok Technological Co., Ltd. (Shenzhen, China) was selected as dispersant.

### 2.2. Experimental Process

The experimental process was basically similar to the previous work [25]. First, uniform Ni-Ti blended powders with 1:1 atomic ratio were mixed by ball milling for 2 h with the protection of Ar atmosphere at a rate of 120 rpm and with the weight ratio of ball to powders of 1:1. Meanwhile, the monomer HEMA and crosslinker HDDA were dissolved into the solvent to prepare a premixed solution with a concentration of 30 wt.%. Subsequently, the blended powders and dispersant silok-7456F were added to the premixed solution and milled at a rate of 120 rpm for 12 h to prepare homogeneous slurries with the solid loading of 50 vol.%. Next, a little initiator and catalyst were injected into the slurries. After using a glass rod to stir well rapidly, the slurries were filled into a PTFE mold and quickly put in an electric drying oven at 70 °C for curing to form wet green compact. After demolding, it was rapidly heated to 180 °C for 60 min and 450 °C for 90 min in a tubular furnace under the protection of Ar gas to remove the organic compounds completely. Finally, the debinded samples were placed into a microwave sintering furnace (HY-SZ4516, Huaye Co., Ltd., Changsha, China) and heated to 950 °C, 1000 °C, 1050 °C and 1100 °C at a heating rate of 20 °C/min for 30 min to prepare porous NiTi specimens. A flowing Ar atmosphere with high purity was used in the chamber to avoid the oxidation of metal powder during the sintering.

### 2.3. Characterization

The porosities of the sintered specimens were measured by a hydrometer (DE-120M, Daho Meter, Dongguan, China) based on Archimedes’ principle. The microstructures of sintered samples were observed by POLYVAR-METHMV-2 metallurgical microscope, and the pore size distribution was obtained by dealing with the metallographic images using Image-Pro software. The micromorphology of sintered samples and powders was observed with a scanning electron microscope (SEM, MIRA3, TESCAN, Brno, Czech Republic). The compressive strength was tested under the electronic universal testing machine (DDL-300, Sinotest Equipment, Changchun, China) using the Φ6 mm × 9 mm specimen, and the loading rate was 0.5 mm/min. The phase composition of the sintered specimen was analyzed by an X-ray diffractometer (XRD, D/Max2500, Rigaku, Tokyo, Japan) with Cu-Kα radiation (λ = 0.154 nm).

## 3. Results and Discussion

### 3.1. Pore Structure

The microstructures of the porous NiTi alloy prepared at different sintering temperature are exhibited in Figure 2, and the corresponding pore size distributions are presented in Figure 3.

As seen in Figure 2, there were numerous pores with different sizes and shapes distributed in the specimen. A three-dimensionally interconnected pore structure was formed and adjacent grains were interconnected to form a strong skeleton. The pores with a large size are directly inherited from the original organic reagent and solvent in slurry, and the pore size is reflects the space between uniformly dispersed particles. In contrast, the small pores distributed in the pore wall are mainly derived from the Kirkendall effect caused by the different diffusion velocity between Ti and Ni atoms, which will cause the formation of pores in the nickel side [26,27]. Meanwhile, as the temperature increased, the pore size, number and connectivity of pores are reduced gradually. When the sintering temperature reached 1100 °C, a few closed pores were observed. During this process, the pore shape became more and more regular. From Figure 3, it was found that the pore distribution reduced from 0–90 μm to 0–50 μm as the temperature increased from 950 °C to 1100 °C, which indicated the maximum pore diameter gradually decreased. Thus, the homogeneity of pores is improved with the rise of sintering temperature. Figure 4 summaries the porosity and the mean pore diameter data.

As shown in Figure 4, both the porosity and the mean pore diameter showed a downward trend with the rise of sintering temperature. When the temperature was raised from 950 °C to 1100 °C, the porosity of the sintered specimen would decrease from 40.68% to 35.41% and the mean pore diameter would decrease from 30.07 μm to17.65 μm. The results coincide with the microstructure evolution of porous NiTi alloy in Figure 1. The essence of sintering is the process of element diffusion, alloying and particle densification. The higher the temperature is, the greater the activity of the element is, and its alloying and densification rate will increase, which will lead to the shrinkage of the sample and the reduction of the porosity and pore size. Based on the phase diagram of Ti-Ni, when the temperature exceeds the eutectic temperature of 942 °C, a transient liquid (titanium rich) forms which will intensify the change [27]. Once the temperature exceeds 1050 °C, the driving force of sintering is reduced due to the sharp reduction of free energy, and further increase in temperature has little effect on the change of porosity. According to Ref. [28], a porosity in the range of 30 to 50% is appropriate for the ingrowth of capillaries and new bone tissue. Therefore, the pore characteristics of porous NiTi alloy fabricated can well match those of bone. The results also suggest that the pore structure of materials by gel-casting can be tailored by adjusting the sintering temperature.

### 3.2. Phase Composition

Figure 5 shows the XRD patterns of Ni-Ti blended powders and sintered specimens at various temperatures.

The Ti powder used in the experiment was obtained by the method of hydrogenation dehydrogenation and the particle size was small (mean diameter of 6.37 μm), so there was TiD_1.5_ phase in Ni-Ti blended powders. In Figure 5b, NiTi, Ni_3_Ti and Ti_2_Ni phases were identified, which illustrated Ni and Ti powders have a reaction during sintering densification. The presence of TiO_2_ phase was the result of oxygen contamination from the organic reagent or protective atmosphere. As can be seen from Figure 5, the temperature has an important effect on the phase composition of the sintered specimen. At the temperature of 950 ℃, the predominant phases were Ni_3_Ti and Ti_2_Ni, and the pure Ni peak was also being detected. The insufficient atomic diffusion between the Ni and Ti at low temperature may be responsible for the phenomenon. During heating, the following chemical diffusion reactions take place between nickel and titanium powders [29]:(1)Ni+Ti→NiTi+67kJ/mol
(2)$$\mathrm{Ni}+2\mathrm{Ti}\to {\mathrm{Ti}}_{2}\mathrm{Ni}+83\mathrm{kJ}/\mathrm{mol}$$
(3)$$3\mathrm{Ni}+\mathrm{Ti}\to {\mathrm{Ni}}_{3}\mathrm{Ti}+140\mathrm{kJ}/\mathrm{mol}$$

Based on the phase diagram of Ti-Ni, NiTi_2_, Ni_3_Ti and NiTi are stable compounds in the Ti-Ni system [27]. Moreover, according to thermodynamic principles, reactions (2) and (3) take precedence over reaction (1). Therefore, Ni_3_Ti and Ti_2_Ni phases are formed preferentially. As the sintering temperature rose, the NiTi phase gradually increased and became the predominant phase. At the same time, no nickel signal was detected, which can be ascribed to the enhancement of interdiffusion between nickel and titanium atoms due to the higher temperature. When the temperature reached 1100 °C, the diffraction peak intensity of NiTi phase was far higher than that of Ti_2_Ni phase. Obviously, it is hard to get a porous NiTi alloy with a single NiTi phase through diffusion of nickel and titanium powder, and the results are almost consistent with the Refs. [12,13,19,26,30]. Compared with our previous work [31], the relative strength of the NiTi peak of the microwave sintered specimen was higher than that of conventional sintering at the same temperature and shorter holding time. This suggests that microwave action during densification may have contributed to diffusion of the atoms.

### 3.3. Mechanical Properties

Figure 6 and Figure 7 show the compressive stress-strain curves of sintered specimen prepared at various temperatures and their corresponding compressive strength and elastic modulus, respectively.

Figure 6 reveals that the sintered specimen under different temperatures presented similar mechanical behavior, and the compressive stress-strain curves can be divided into the following three stages [32,33]: (1) Linear elastic deformation stage, where the pore wall responded to elastic compression and the stress increased linearly with the strain. (2) Plastic yield deformation stage, where plastic deformation occurred preferentially around the micropores due to stress concentration, and then extended to the whole pore wall accompanied by rapid crack propagation. (3) Densification and rupture stage, where the pore walls would collapse and the rupture of the specimen took place. It was also found that the maximum stress strength and the maximum strain were promoted dramatically with the rise in temperature. As shown in Figure 7, as the temperature rose, both the compressive strength and elastic modulus of the sintered porous specimen gradually improved. When the sintering temperature was raised from 950 °C to 1100 °C, they increased from 185 MPa and 3.45 GPa to 390 MPa and 6.9 GPa, respectively. It is widely known that the mechanical properties of porous materials depend on their microstructure, such as pore size and shape, porosity, the densification degree of the pore wall and the phase composition. Particularly, the sharp edges of the irregular pores are prone to stress concentration, which will lead to the deterioration of the strength and plasticity of the porous materials. It can also be found from Figure 2 and Figure 5, that with the rise in temperature, the relative density and the content of NiTi phase and the densification degree of the pore wall increased, while the pore shape became more regular. All of these will improve the strength of sintered porous NiTi. In additional, the elastic modulus of the porous NiTi alloys within 3.45–6.9 GPa may meet the requirement of some natural bone for elastic modulus (3–20 GPa for cortical bone) [34,35].

### 3.4. Fracture

The fracture morphology of porous NiTi specimens prepared under the different temperatures is shown in Figure 8.

Obviously, the sintered porous NiTi specimen presented a typical brittle fracture, which was due to the existence of many pores. It can be seen in Figure 8a that the specimen was not sufficiently sintered at 950 °C, and the poor diffusion and necking among atoms weaken the bonding strength among particles and therefore gave rise to poor mechanical properties. The fracture surface showed that there were some voids between the undeformed and packed particles, suggesting that the fracture probably was a simple powder particle segregation. As the sintering temperature rose to 1100 °C (Figure 8d), the bonding strength among the particles was improved with the growth of sintering neck, implying a stronger resistance to fracture deformation. The number and size of pores in the specimen were also reduced. The fracture surface showed smooth cleavage facets and river patterns, suggesting that fracture emerges by stress concentration around the pores and separation of grains [36]. As a result, the fracture mechanical properties may be improved at increased sintering temperature.

## 4. Conclusions

Porous NiTi alloys were fabricated by gel-casting combined with microwave sintering, using HEMA-HDDA as a gel system. The influence of sintering temperature on the microstructure and mechanical properties was studied. The following conclusions can be drawn:(1)With increasing sintering temperature, the atoms diffused sufficiently and the morphology of pores became smooth and regular. Moreover, the porosity and the mean pore diameter of the specimen was correspondingly reduced.(2)With increasing sintering temperature, the compressive strength and elastic modulus of the sintered porous NiTi specimen gradually improved.(3)The sintered porous NiTi alloy was composed of the major NiTi phase with a few Ti_2_Ni and Ni_3_Ti second phases, and the relative intensity of the NiTi phase increased with the temperature. It presented a typical brittle fracture.(4)The pore structure and mechanical properties of samples can be tailored by controlling sintering temperature. A porous NiTi alloy with the porosity of 38.9%, the compressive strength of 254 MPa and the elastic modulus of 4 GPa was obtained at the sintering temperature of 1000 °C for 30 min, which could be in accord with the demand for bone replacement.

## Figures and Tables

**Figure 1 materials-15-07331-f001:**
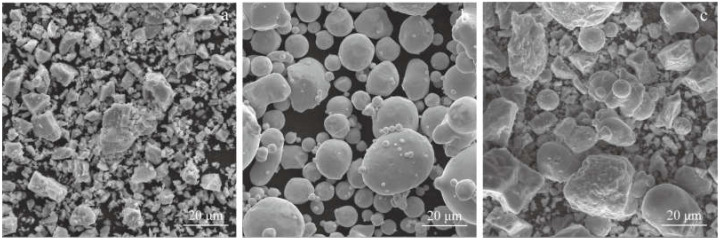
SEM images of the powders: (**a**) Ti powder; (**b**) Ni powder; (**c**) Ni-Ti blended powders.

**Figure 2 materials-15-07331-f002:**
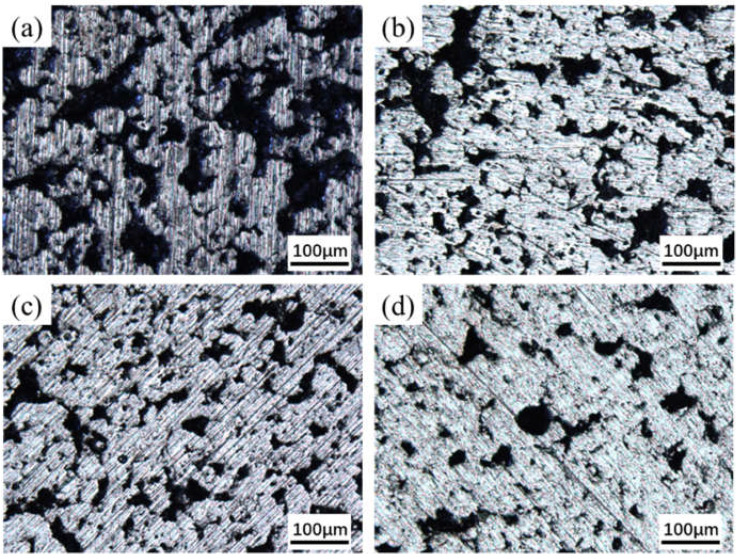
Optical micrographs of porous NiTi alloys prepared at various temperatures: (**a**) 950 °C; (**b**) 1000 °C; (**c**) 1050 °C; (**d**) 1100 °C.

**Figure 3 materials-15-07331-f003:**
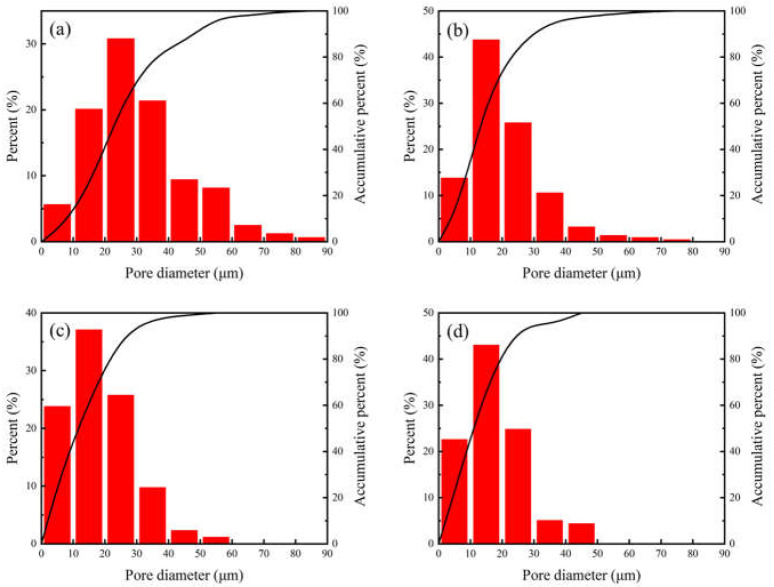
The relation between pore size distribution of sintered specimen with sintering temperature: (**a**) 950 °C; (**b**) 1000 °C; (**c**) 1050 °C; (**d**) 1100 °C.

**Figure 4 materials-15-07331-f004:**
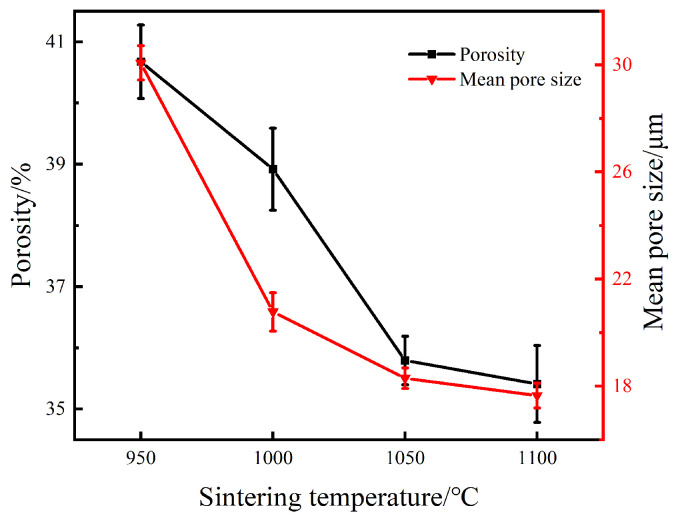
The relation of porosity and average pore size with sintering temperature.

**Figure 5 materials-15-07331-f005:**
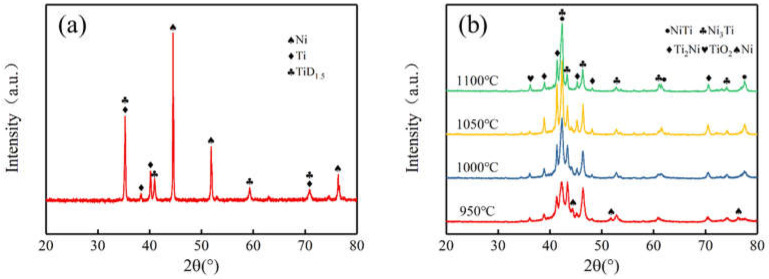
XRD patterns: (**a**) Ni-Ti blended powders; (**b**) porous NiTi specimen.

**Figure 6 materials-15-07331-f006:**
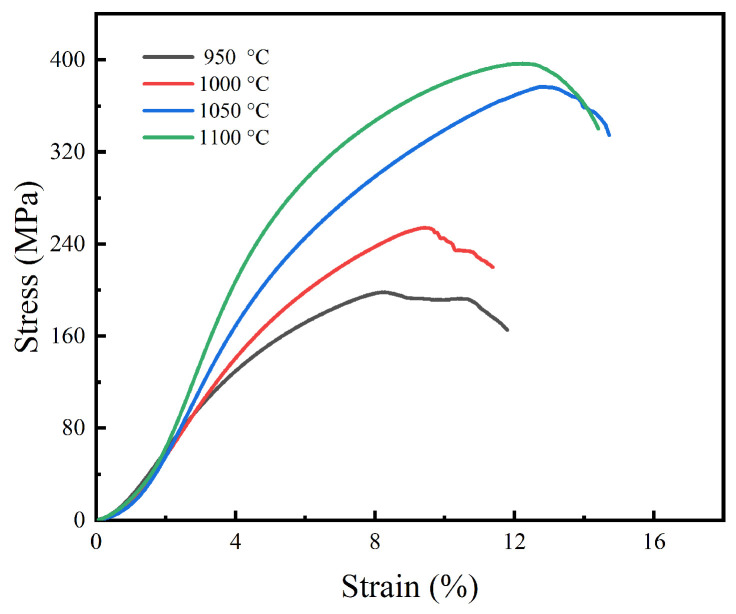
The compressive behavior of porous NiTi specimen.

**Figure 7 materials-15-07331-f007:**
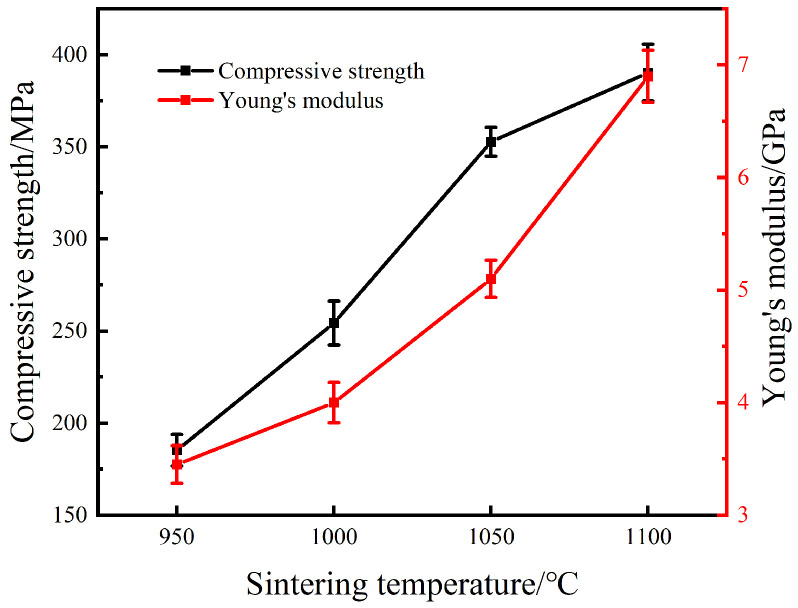
The relation of the compressive strength and elastic modulus of the specimen with its sintering temperature.

**Figure 8 materials-15-07331-f008:**
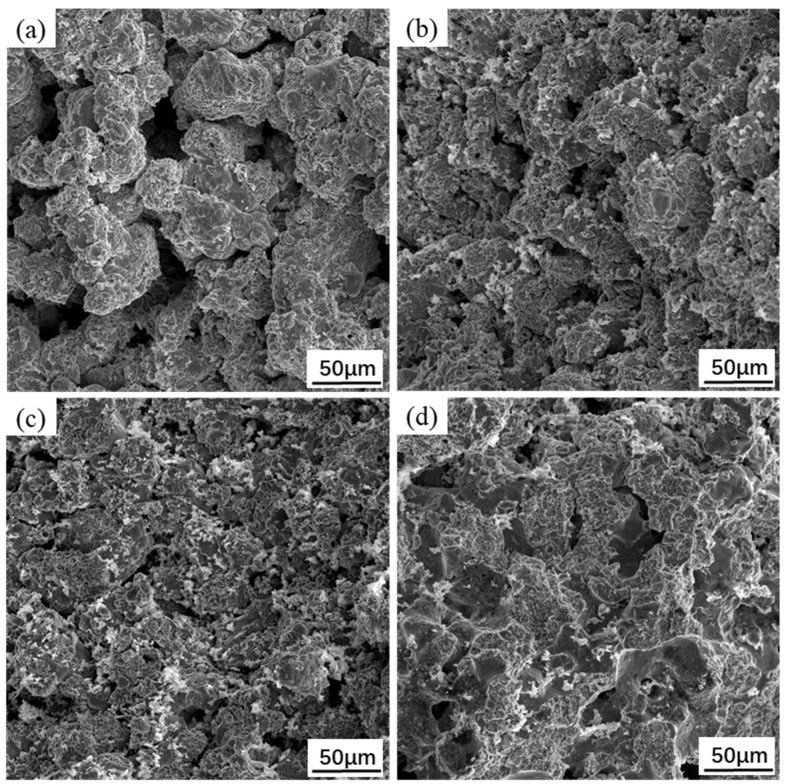
Fracture morphology of sintered porous NiTi specimen at various temperatures: (**a**) 950 °C; (**b**) 1000 °C; (**c**) 1050 °C; (**d**) 1100 °C.

## Data Availability

All the data available in main text.

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
