# Peer review of "Microstructure and Mechanical Properties of Porous NiTi Alloy Prepared by Integration of Gel-Casting and Microwave Sintering"

_materials, 2022, doi:10.3390/ma15207331_

Round 1
Reviewer 1 Report
The article «Microstructure and mechanical properties of porous NiTi alloy prepared by integration of gel-casting and microwave sintering» is well written and the topic is relevant. The results obtained are in good agreement with the results of conventional diffusion sintering of TiNi powder. I recommend accepting the article after eliminating a number of comments.

Reviewer 2 Report
Zhiqiang et al have submitted an interesting, well-structured manuscript about the combination of sol-gel casting and microwave sintering. I only recommend small changes before publish it. (My "true" recommendation is between "Accept in present form" and "Accept after minor revision".)
1. Line 70: just "First,..."
2. Section 2.2.: a figure about the experimental process would be welcomed, but is not strictly necessary.
3. Section 2.3.: "X-ray" (with capital letter). What was the X-ray source (what was the wavelength of the radiation)?
4. Figure 2.: micron markers should be more visible.
5. Line 121. about that the distribution became more narrow. Could you quantify it eg. with the standard deviation?
6. Figure 4 and line 130.: please indicate the margins of errors (at least in the figure).
7. Figure 7 and line 195.: please indicate the margins of errors (at least in the figure).
8. Through the whole document: please check all the temperature values be presented in the correct form "xxx °C".
Reviewer 3 Report
1. Some of the sentences in the abstract is repeated as it is in the introduction text as well. Revise it.
2. Grammatical and typo errors must be revised.
3. During ball milling of NiTi, there is always a greater chance of oxidation of Titanium. How did authors managed to get rid of oxidation?
4. It is recommended to add the SEM image after ball milling to understand the morphology of the powders before sintering.
5. It is recommended to revise the sentence “When the temperature raised from 950℃ to 1100℃, the porosity and the mean pore diameter of sintered specimen would vary from 40.68%, 30.07 μm to 35.41%, 17.65 μm, respectively”. It is confusing to the readers.
6. In figure 5 XRD, It is better to add even the XRD of ball milled powder.
